# Clinical Significance of the Neutrophil–Lymphocyte Ratio as an Early Predictive Marker for Adverse Outcomes in Patients with Acute Cholangitis

**DOI:** 10.3390/medicina58020255

**Published:** 2022-02-09

**Authors:** Sang-Hoon Lee, Tae-Yoon Lee, Jong-Hyeon Jeong, Young-Koog Cheon

**Affiliations:** Department of Internal Medicine, Konkuk University School of Medicine, Seoul 05030, Korea; lshjjang_2000@hanmail.net (S.-H.L.); 20160091@kuh.ac.kr (J.-H.J.); yksky001@hanmail.net (Y.-K.C.)

**Keywords:** acute cholangitis, prognosis, neutrophil–lymphocyte ratio

## Abstract

*Background and objectives:* Acute cholangitis can be life-threatening if not recognized early. We investigated the predictive value of the neutrophil–lymphocyte ratio (NLR) in acute cholangitis. *Materials and Methods:* We retrospectively evaluated 206 patients with acute cholangitis who underwent biliary drainage. The severity of acute cholangitis was graded according to the Tokyo 2018 guideline. Patients were dichotomized according to the acute cholangitis severity (mild/moderate vs. severe), the presence of shock requiring a vasopressor/inotrope, and blood culture positivity. The baseline NLR, white blood cell (WBC) count, and C-reactive protein (CRP) levels were compared between groups. *Results:* The severity of acute cholangitis was graded as mild, moderate, or severe in 71 (34.5%), 107 (51.9%), and 28 (13.6%) patients, respectively. Ten patients (4.8%) developed shock. Positive blood culture (*n* = 50) was observed more frequently in severe acute cholangitis (67.9% vs. 17.4%, *p* < 0.001). The NLR was significantly higher in patients with severe cholangitis, shock, and positive blood culture. The area under the curve (AUC) for the NLR, WBC, and CRP for severe acute cholangitis was 0.87, 0.73, and 0.74, respectively. The AUC for the NLR, WBC, and CRP for shock was 0.81, 0.64, and 0.67, respectively. The AUC for the NLR, WBC, and CRP for positive blood culture was 0.76, 0.64, and 0.61, respectively; the NLR had greater power to predict disease severity, shock, and positive blood culture. The optimal cut-off value of the baseline NLR for the prediction of severe acute cholangitis, shock, and positive blood culture was 15.24 (sensitivity, 85%; specificity, 79%), 15.54 (sensitivity, 80%; specificity, 73%), and 12.35 (sensitivity, 72%; specificity, 70%), respectively. The sequential NLR values from admission to 2 days after admission were significantly higher in patients with severe cholangitis and shock. *Conclusions:* An elevated NLR correlates with severe acute cholangitis, shock, and positive blood culture. Serial NLR can track the clinical course of acute cholangitis.

## 1. Introduction

Acute cholangitis is an acute inflammatory disease of the bile duct that occurs when biliary obstruction results in cholestasis and biliary infection [1]. Biliary obstruction increases the pressure within the bile duct and bacteria or endotoxins migrate from the bile into the systemic circulation [2]. Acute cholangitis has a broad spectrum of severity, from mild conditions that resolve with conservative treatment, to severe cases progressing to septic shock [3]. The majority of acute cholangitis cases have a favorable outcome. However, in severe acute cholangitis, failure to recognize the severity of the disease and start early management leads to high morbidity and mortality rates. The proper use of antibiotics and biliary drainage has reportedly decreased the mortality rate for acute cholangitis. Thus, the early prediction of severity in acute cholangitis is important for emergency treatment.

Several studies have investigated predictors of acute cholangitis severity, suggesting procalcitonin, presepsin, IL-7, and the delta neutrophil index (DNI) as prognostic markers [3,4,5,6,7,8]. However, their high cost or dependence on specific analyzers prevents their wide availability. There is a need for new, inexpensive, and simple predictors of severe acute cholangitis, to stratify patients for urgent biliary drainage, close monitoring, and selective transfer to the intensive care unit (ICU) as early as possible.

The neutrophil–lymphocyte ratio (NLR) is reportedly predictive of adverse outcomes in acute pancreatobiliary diseases [9,10,11,12,13]. The NLR reflects the immune and inflammatory responses more precisely than the total white blood cell (WBC) count [9]. Clinical research confirmed the sensitivity of the NLR for the diagnosis/stratification of systemic infection, sepsis, and bacteremia, as well as its robust predictive and prognostic value [14]. The NLR is predictive of disease severity, septic shock, organ failure, or ICU admission in acute pancreatitis and cholecystitis [9,12,15]. However, no study has investigated the association of the NLR with severe acute cholangitis. We evaluated the utility of the NLR for predicting the severity of acute cholangitis. 

## 2. Materials and Methods

### 2.1. Patients and Data Collection

We retrospectively evaluated 206 patients with acute cholangitis who underwent biliary drainage between 1 April 2018 and 30 November 2021, at Konkuk University Medical Center, Seoul, Korea. The method and timing of biliary decompression (endoscopic retrograde cholangiopancreatography (ERCP) or percutaneous transhepatic biliary drainage (PTBD)) were at the discretion of the attending physician. Demographic (age, sex, and the etiology of acute cholangitis), clinical, and procedural data were obtained from the electronic medical records. The clinical data comprised the results of blood culture and laboratory values (WBC count, NLR, platelet count, prothrombin time international normalized ratio (PT-INR), C-reactive protein (CRP), total bilirubin, albumin, and creatinine), which were collected on admission. A Sequential Organ Failure Assessment (SOFA) score, previously known as the sepsis-related organ failure assessment score, was determined on admission [16]. The procedural data comprised the timing and method of biliary drainage.

All patients received intravenous antibiotics with a third-generation cephalosporin, piperacillin/tazobactam, ciprofloxacin or carbapenem, immediately after the diagnosis of acute cholangitis. Fluid replacement was performed with crystalloid solutions. In cases of hypotension even after appropriate hydration, an inotropic agent was given. Intravenous antibiotics were administered until the inflammatory signs resolved.

All the patients enrolled in the current study fulfilled the diagnostic criteria for acute cholangitis according to the Tokyo guideline 2018 (TG18) [17]. The diagnostic criteria were as follows: (1) systemic inflammation (fever (≥38 °C) or evidence of an inflammatory response (WBC count > 10,000/µL or <4000/µL, or CRP level ≥ 1 mg/dL)); (2) cholestasis (serum total bilirubin ≥ 1 mg/dL or abnormal liver function test (alkaline phosphatase, gamma-glutamyl transferase, aspartate aminotransferase, and alanine aminotransferase > 1.5 × upper limit of normal value (ULN))); and (3) biliary dilatation or evidence of the etiology on imaging. We excluded patients who had concurrent acute inflammatory diseases (e.g., acute biliary pancreatitis, liver abscess, pneumonia, or histologic diagnosis of acute cholecystitis in subsequent cholecystectomy) at admission and during hospitalization because these can confound the association between the NLR and acute cholangitis. We excluded patients less than 18 years of age, those who experienced early adverse events associated with ERCP or PTBD (e.g., pancreatitis, perforation, bleeding, or biloma), and those with missing data. We also excluded patients who had received chemotherapy or anticoagulants because these frequently cause thrombocytopenia, neutropenia, or PT prolongation. Liver transplant recipients were not included because liver transplantation is not performed in our hospital. This study was approved by the Institutional Review Board of the hospital. Informed consent was waived by the board.

### 2.2. Study Outcomes and Definitions

The primary outcome was whether the NLR on admission could predict the severity of acute cholangitis, shock requiring a vasopressor/inotrope, and positive blood culture. The secondary outcome was the optimal NLR cut-off value for predicting disease severity, shock, and blood culture positivity.

The severity of acute cholangitis was classified as follows according to the TG18 [17]. Severe acute cholangitis was defined if at least one dysfunction was noted in the cardiovascular, neurological, respiratory, renal, hepatic, or hematologic systems. Moderate acute cholangitis was defined as any two of the following conditions: (1) abnormal WBC count (>12,000/µL or <4000/µL), (2) high fever (≥39 °C), (3) advanced age (≥75), (4) hyperbilirubinemia (total bilirubin ≥ 5 mg/dL), and (5) hypoalbuminemia (serum albumin < ULN × 0.7). Mild acute cholangitis was defined as not meeting the criteria for moderate or severe acute cholangitis. Shock assessed at admission and during hospitalization was defined as hypotension requiring dopamine ≥5 μg/kg or any dose of norepinephrine.

The patients with acute cholangitis were classified according to the disease severity (mild-to-moderate vs. severe acute cholangitis), the presence of shock requiring a vasopressor/inotrope, and positive blood culture. The association of the NLR with severe acute cholangitis, shock, and positive blood culture was investigated. The NLR was determined by calculating the ratio between the absolute neutrophil and lymphocyte counts, and the NLR values were calculated at the time of hospitalization and on day 1 and day 2 after admission. The drainage time was defined as the time (hours) from the hospital visit to receiving biliary drainage procedures such as ERCP or PTBD. Early complications associated with ERCP or PTBD were defined as complications that occurred within 30 days of ERCP or PTBD.

### 2.3. Statistical Analysis

We used SPSS for Windows 28.0 software (SPSS Inc., Chicago, IL, USA) and MedCalc 20.027 (MedCalc Software, Ostend, Belgium) for the statistical analysis. Descriptive statistics for continuous and categorical variables are presented as the mean ± standard deviation and values (%), respectively. The differences between categorical variables were analyzed using the chi-square test and Fisher’s exact test, and those between continuous variables with a normal distribution using the Student’s *t*-test, as appropriate. We checked the normality in the distribution of the quantitative data with the Kolmogorov–Smirnov test. Skewed distribution variables were described as medians with interquartile range (IQR) and the Mann–Whitney U test was used to compare them. The NLR was treated as a continuous variable, and the distributions of acute cholangitis severity, shock, and positive blood culture were treated as categorical variables. A receiver operating characteristic (ROC) curve was plotted, and the area under the curve (AUC) was calculated to determine the optimal cut-off NLR value; 95% confidence intervals (CI) were also obtained. A *p*-value < 0.05 indicated statistical significance.

## 3. Results

### 3.1. Clinical Features According to Acute Cholangitis Severity and Isolated Pathogens

Among 300 patients with acute cholangitis who were hospitalized, 206 satisfied the inclusion criteria and 94 were excluded (Figure 1). 

Of the 206 acute cholangitis patients analyzed, 71 (34.5%) had mild, 107 (51.9%) had moderate, and 28 (13.6%) had severe acute cholangitis. Among the 28 patients with severe acute cholangitis, 24 had one organ/system dysfunction and 4 had two dysfunctions: hematologic dysfunction in 12, cardiovascular dysfunction in 10, renal dysfunction in 4, hepatic dysfunction in 3, respiratory dysfunction in 2, and neurologic dysfunction in 1. Seven patients (3.3%) were transferred to the ICU, and three patients (1.4%) died within 30 days of hospitalization. The demographic and clinical features of the patients are listed in Table 1.

The mean age of the patients was 73.6 ± 13.4 years and 115 (55.8%) were male. Although the timing of biliary drainage was not different between both groups, PTBD was more frequently performed in the severe acute cholangitis group than the mild/moderate group (39.3% vs. 9%, *p* < 0.001). The platelet, PT-INR, creatinine, and albumin levels were significantly different, according to the severity of acute cholangitis. The hospitalization duration in the severe acute cholangitis group was longer than in the mild/moderate acute cholangitis group (median, 12 vs. 6.5 days, *p* = 0.001). Patients with severe acute cholangitis exhibited a significantly higher SOFA score than patients with mild/moderate acute cholangitis (4 (IQR, 4–6) vs. 2 (IQR, 2–3); *p* < 0.001). Regarding the inflammatory markers, the severe acute cholangitis group had significantly higher WBC (15,600 vs. 9655, *p* < 0.001), CRP (10.15 vs. 4.68, *p* = 0.001) and NLR (22.27 vs. 8.07, *p* < 0.001) than the mild/moderate acute cholangitis group.

Positive blood culture (*n* = 50) was more frequently observed in the severe acute cholangitis group than the mild/moderate cholangitis group (67.9% vs. 17.4%, *p* < 0.001). Fifty-four pathogens were isolated from the blood cultures of 50 patients (Table 2). *Escherichia coli* (*n* = 33, 61.1%) was the most common pathogen of Gram-negative bacteria and *Enterococcus* species (*n* = 5, 9.2%) were the most common pathogens of Gram-positive bacteria. The baseline NLR between Gram-negative and Gram-positive bacterial groups was not significantly different (median, 23.06 vs. 18.32, *p* = 0.718).

### 3.2. Role of NLR, WBC and CRP in Predicting Severe Acute Cholangitis

The ROC curves for the NLR, WBC, and CRP as predictors of severe acute cholangitis are shown in Figure 2. Although all three markers were predictive for severe acute cholangitis (*p* < 0.001 for all), the AUC of the NLR in predicting severe acute cholangitis was 0.87 (95% CI, 0.82–0.92), which was higher than that of the WBC count (0.73 (95% CI, 0.6–0.86)) and the CRP level (0.74 (95% CI, 0.65–0.84)). The optimal cut-off values of the NLR, WBC and CRP for predicting severe acute cholangitis were 15.24 (sensitivity, 85%; specificity, 79%), 11,150 (sensitivity, 71%; specificity, 68%), and 7.18 (sensitivity, 67%; specificity, 66%), respectively.

### 3.3. Role of NLR, WBC, and CRP in Predicting Shock and Positive Blood Culture

There were ten patients (4.8%) with shock requiring a vasopressor/inotrope in the severe acute cholangitis group. The patients with shock had a significantly higher NLR (median, 19.27 vs. 8.66, *p* = 0.02) than patients without shock, but the WBC count (15,665 vs. 9900, *p* = 0.1), and the CRP levels (7.99 vs. 5.07, *p* = 0.097) were not significantly different. The ROC curves showed that the NLR was predictive for severe acute cholangitis (*p* = 0.001), but WBC and CRP were not (*p* = 0.126 and *p* = 0.069, respectively). The NLR had an AUC of 0.81 (95% CI, 0.73–0.89), which was higher than that of the CRP level (0.64 (95% CI, 0.41–0.87)) and the WBC count (0.67 (95% CI, 0.51–0.82)) (Figure 3). The optimal cut-off values of the NLR, WBC and CRP for predicting shock were 15.54 (sensitivity, 80%; specificity, 73%), 10,950 (sensitivity, 60%; specificity, 60%), and 6.92 (sensitivity, 60%; specificity, 60%), respectively. 

The number of cases with positive blood cultures was 50 out of 206 (24.2%); 31 (17.4%) in mild/moderate acute cholangitis and 19 (67.9%) in severe cholangitis (*p* < 0.001). The patients with positive blood cultures had a significantly higher NLR (median, 17.94 vs. 7.83, *p* < 0.001), WBC (11,550 vs. 9625, *p* = 0.002), and CRP (9.3 vs. 6.67, *p* = 0.009) compared to those with negative blood cultures. The AUCs for predicting positive blood culture were 0.76 (95% CI, 0.69–0.84, *p* < 0.001), 0.64 (95% CI, 0.55–0.74, *p* = 0.002) and 0.61 (95% CI, 0.51–0.7, *p* = 0.024) for NLR, WBC, and CRP, respectively (Figure 4); the NLR has greater power to predict positive blood culture than WBC or CRP. The optimal cut-off values of the NLR, WBC and CRP for predicting positive blood culture were 12.35 (sensitivity, 72%; specificity, 70%), 10,890 (sensitivity, 60%; specificity, 64%), and 6.53 (sensitivity, 60%; specificity, 63%), respectively.

### 3.4. Role of SOFA Score in Predicting Severe Acute Cholangitis, Shock, and Positive Blood Culture

The ROC curves for the SOFA score as predictors of severe acute cholangitis, shock, and positive blood culture are shown in Figure 5. The AUCs for predicting severe cholangitis, shock, and positive blood culture were 0.87 (95% CI, 0.82–0.91, *p* < 0.001), 0.93 (95% CI, 0.88–0.96, *p* < 0.001) and 0.72 (95% CI, 0.65–0.78, *p* < 0.001), respectively. The optimal cut-off values of the SOFA score for predicting severe acute cholangitis, shock, and positive blood culture were 3 (sensitivity, 89%; specificity, 84%), 3 (sensitivity, 100%; specificity, 78.1%), and 2 (sensitivity, 82%; specificity, 52%), respectively.

### 3.5. Changes over Time in the NLR According to Severity of Acute Cholangitis and Presence of Shock

The sequential NLR values in the mild/moderate and severe acute cholangitis groups are listed in Table 3. The severe cholangitis group had a significantly higher NLR than the mild/moderate group on all 3 days (median, 22.27, 12.11, and 9.63 vs. 8.07, 5.38, and 3.48, respectively, *p* < 0.001). Although both groups showed the highest NLR on admission with a gradual reduction after 24 h of hospitalization, a decrease close to the normal range (NLR of 1–3) [18] was found at 2 days after admission in the mild/moderate cholangitis group.

We evaluated the differences in the NLRs by grouping the patients with acute cholangitis into those with and without shock. All of the baseline NLR values and those on day 1 and 2 were significantly higher in patients with shock than in patients without shock (median, 19.27, 13.75, and 12.57 vs. 8.66, 5.69, and 3.76, *p* < 0.05) (Table 4). Although both groups showed the highest NLR on admission followed by a gradual reduction thereafter, the NLR on day 2 in patients with shock remained higher than the highest score in patients without shock (12.57 vs. 8.66), indicating ongoing inflammation. A decrease in the NLR close to the normal range was noted on day 2 after hospitalization in patients without shock.

## 4. Discussion

We evaluated whether the NLR can predict severe acute cholangitis. We found that the NLR increased significantly according to the severity of acute cholangitis, the presence of shock requiring a vasopressor/inotrope, and positive blood culture. In the ROC analysis, the NLR could discriminate severe acute cholangitis (AUC 0.87), shock (0.81), and positive blood culture (0.76) better than WBC or CRP. The cut-off values of 15.24 and 15.54 showed sensitivities of 85% and 80%, and specificities of 79% and 73% for predicting severe acute cholangitis and shock. Additionally, we investigated changes in the NLR value over time and demonstrated that NLR values for all 3 days during hospitalization were significantly higher in patients with severe acute cholangitis and shock than those without severe cholangitis and shock. A SOFA score ≥3 was also significantly associated with severe acute cholangitis and shock.

Since Zahorec et al. observed a marked elevation of neutrophil counts and a deep decline in lymphocyte counts in patients with critical illness [19], the NLR has been studied in various infectious and non-infectious diseases [20,21]. The higher the NLR, the more unbalanced the immune–inflammatory status, because the NLR reflects the balance of neutrophilia and lymphopenia, through the combined response of endogenous cortisol and catecholamines. Acute cholangitis is associated with bacterial translocation from the gut and elevated bilirubin levels, which play critical roles in innate immunity activation [7]. Neutrophil is an important component of the innate immune system during sepsis, and promotes systemic inflammation by secreting proinflammatory cytokines, leading to SIRS and septic shock in severe acute cholangitis [8]. Lymphocytes regulate systemic inflammation as the disease progresses. Ongoing inflammation causes lymphopenia as a consequence of lymphocyte redistribution and apoptosis. Therefore, the NLR may reflect dynamic changes in the immune system in acute cholangitis. Although a previous study has reported that the NLR had superior discriminative power to WBC in the diagnosis of acute cholangitis [22], the usefulness of the NLR for predicting adverse outcomes of acute cholangitis has rarely been investigated. Since prompt management, including antibiotics and biliary drainage, can improve the prognosis of severe acute cholangitis, the identification of a biomarker to predict its severity at an early stage enables stratification by severity and the early initiation of treatment.

To date, suggested predictors for the severity of acute cholangitis include procalcitonin, presepsin, IL-7, and DNI. Shinya et al. reported that a high procalcitonin level on admission is an independent predictor of severe disease and positive blood culture in acute cholangitis [4]. Similarly, another Japanese study of 213 patients reported that the AUC for procalcitonin in predicting severe acute cholangitis was 0.9, which was significantly higher than those of WBC (0.62) and CRP (0.7) [5]. Lee et al. analyzed 204 patients with acute cholangitis, of whom 26 had severe disease. Procalcitonin was significantly predictive of severe acute cholangitis, blood culture positivity, and clinical deterioration; the optimal cut-off values were 1.76, 0.68 and 3.77, respectively [3]. A recent study reported that presepsin is superior to procalcitonin in predicting severe acute cholangitis (AUC 0.71 vs. 0.61), and has the same predictive power (0.68) as procalcitonin for positive blood culture [7]. In spite of their promising predictability, serum procalcitonin and presepsin are not routinely measured in all institutions, and their relatively high cost (procalcitonin USD 23, presepsin USD 35, in Korea) prevents serial assessment during hospitalization. Suwa et al. reported that low IL-7 levels were associated with severe acute cholangitis [6]. There is one study examining the association between DNI and the prognosis of acute cholangitis. In a Korean study of 461 patients, high DNI levels on admission and on day 1 and day 2, are predictive markers of shock and 28 day mortality in patients with acute cholangitis [8]. Although DNI is repeatedly measurable due to its low cost, and simultaneously obtainable with routine complete blood counts (CBC), only one company (Siemens) is manufacturing a blood analyzer that measures DNI automatically. Thus, there is a need for an indicator that is cheap and routinely measured in general clinical practice, to predict the severity of acute cholangitis with high accuracy.

Although the TG18 is known to be the standard for the severity assessment of acute cholangitis, this guideline should include the evaluation of six organ systems, requiring complex items such as vital signs and laboratory results, accompanied by diagnostic criteria for acute cholangitis; accordingly, this guideline is not easily applicable to an emergency setting [8]. The NLR is inexpensive (USD 5.5 in Korea), simply measurable in clinical practice, and unaffected by the CBC analyzer. The NLR on admission can be used to identify patients who need close monitoring, urgent biliary drainage, and possible transfer to the ICU in the emergency department. The NLR on days 1 and 2 could track the clinical course of acute cholangitis. The value of the NLR changed dynamically over time after onset, moving close to normal in patients with mild/moderate acute cholangitis, and remaining elevated far from its normal range in those with severe cholangitis. These findings were consistent between patients with and without shock. As positive blood culture suggests high bacterial load, related to a high risk of septicemia and septic shock, it is crucial to predict positive blood culture results [3]. The NLR was superior to the WBC count and CRP levels for predicting blood culture positivity.

This study showed that mortality was 1.5%, which is low compared to the 5–10% in other studies but is similar to those of a Korean study by Park et al. performing early ERCP [23]. The cause of the good outcomes may be due to conducting early biliary drainage within 48 h in most patients with acute cholangitis. We treated 187 of 206 patients (90.7%) with early ERCP or PTBD.

The major strength of our study is that this is the first report of the usefulness of the NLR for predicting acute cholangitis severity, the occurrence of shock, and positive blood culture. Another strength is that we used strict inclusion criteria, excluding patients with concurrent acute biliary pancreatitis on admission, and acute cholecystitis on subsequent cholecystectomy. Although these conditions can confound the association between the NLR and acute cholangitis, previous studies did not exclude these patients [3,4,5,7]. However, this study has several limitations. Firstly, this was a retrospective study with a relatively small number of patients. Furthermore, a sample size of 186 patients in each group will be required to achieve a power of 80% with a confidence level of 5% to confirm these results, when we assumed a 7.5% incidence rate of mild/moderate acute cholangitis and a 1.5% incidence of severe acute cholangitis among patients with gallstone disease. So, the study may be underpowered in terms of drawing definitive conclusions on the association between the NLR and severe acute cholangitis. Secondly, the study may have certain biases inherently associated with a single-center setting. Thirdly, the results may not generalize to patients with acute cholangitis and concurrent acute inflammatory disease or other procedural complications, because the lymphocyte and neutrophil counts may be confounded by these conditions. Finally, missing data is a limitation in our study.

## 5. Conclusions

In conclusion, the NLR has an ability to predict severe disease, shock requiring a vasopressor/inotrope, and positive blood culture in patients with acute cholangitis, and serial measurement of the NLR can assess change in severity over time. This finding needs to be confirmed in a large prospective trial.

## Figures and Tables

**Figure 1 medicina-58-00255-f001:**
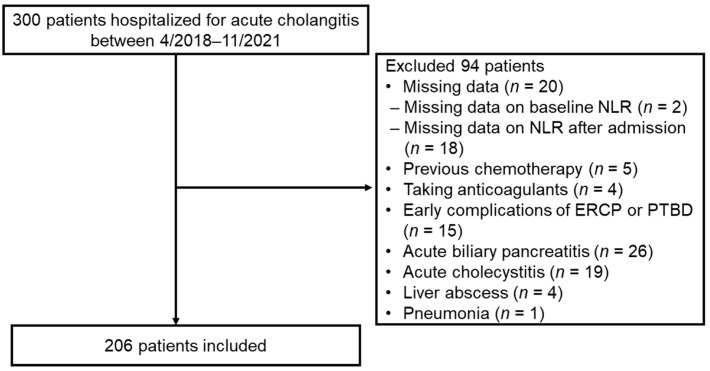
Flowchart of patient enrollment. NLR, neutrophil–lymphocyte ratio; ERCP, endoscopic retrograde cholangiopancreatography; PTBD, percutaneous transhepatic biliary drainage.

**Figure 2 medicina-58-00255-f002:**
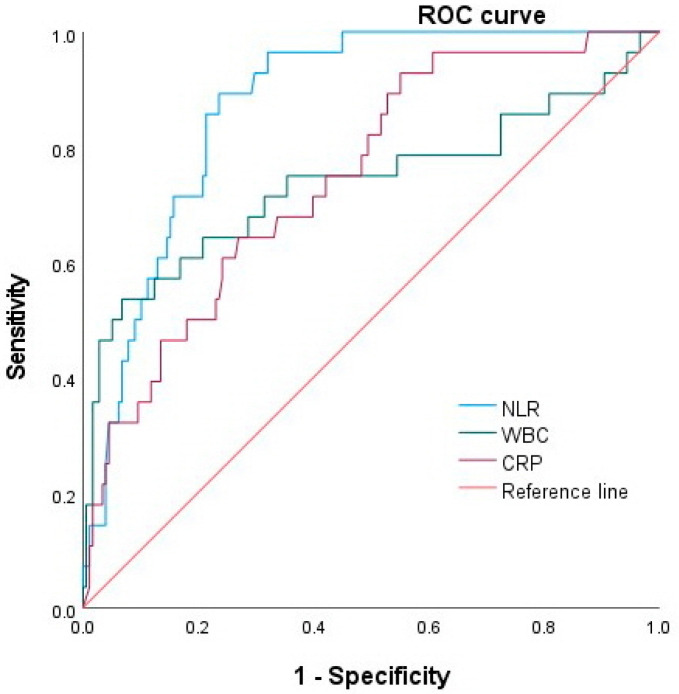
Receiver operating characteristic curve for NLR, WBC and CRP as predictors of severe acute cholangitis. NLR, neutrophil–lymphocyte ratio; WBC, white blood cell; CRP, C-reactive protein.

**Figure 3 medicina-58-00255-f003:**
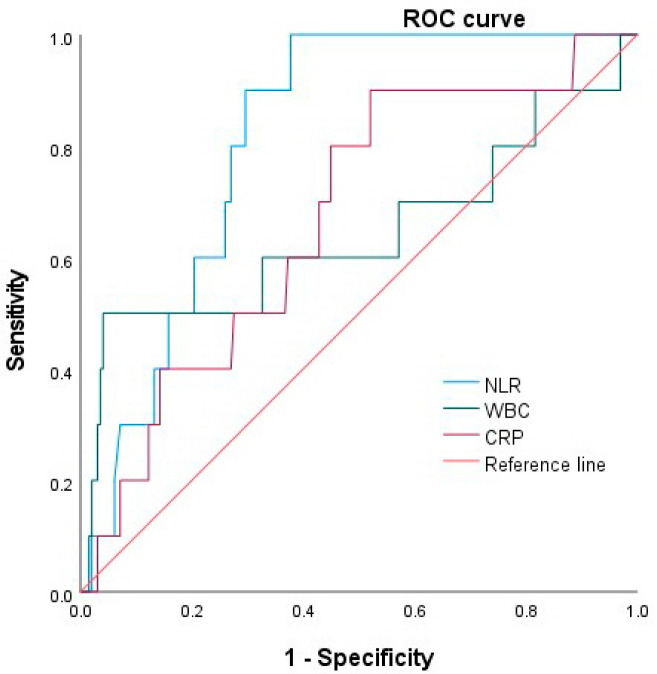
Receiver operating characteristic curves for NLR, WBC and CRP as predictors of shock. NLR, neutrophil–lymphocyte ratio; WBC, white blood cell; CRP, C-reactive protein.

**Figure 4 medicina-58-00255-f004:**
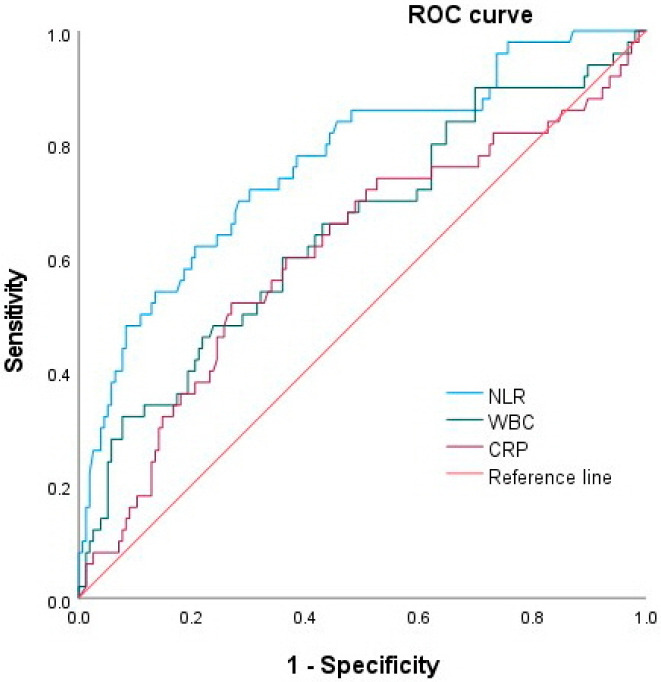
Receiver operating characteristic curve for NLR, WBC and CRP as predictors of positive blood culture. NLR, neutrophil–lymphocyte ratio; WBC, white blood cell; CRP, C-reactive protein.

**Figure 5 medicina-58-00255-f005:**
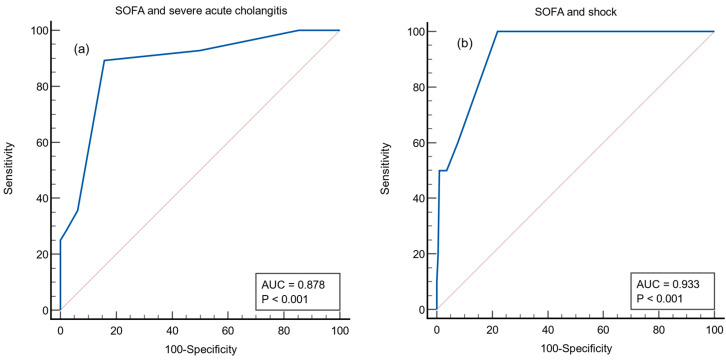
Receiver operating characteristic curve for the SOFA score as a predictor of severe acute cholangitis (**a**), shock (**b**), and positive blood culture (**c**). SOFA, Sequential Organ Failure Assessment.

**Table 1 medicina-58-00255-t001:** Demographic and clinical characteristics of 206 patients diagnosed with acute cholangitis.

	Mild-to-Moderate Acute Cholangitis (*n* = 178)	Severe Acute Cholangitis (*n* = 28)	*p-*Value
Age, mean ± SD, years	72.9 ± 13.6	78.07 ± 11	0.058
Male, gender, *n* (%)	97 (54.5%)	18 (64.3%)	0.414
Etiology of acute cholangitis			
Choledocholithiasis	144 (80.9%)	24 (85.7%)	0.793
Benign biliary stricture	10 (5.6%)	0 (0%)	0.364
Malignant biliary stricture	24 (13.5%)	4 (14.3%)	1.000
Cholangiocarcinoma	16 (9%)	1 (3.6%)	
Pancreatic cancer	1 (0.6%)	0 (0%)	
Gallbladder cancer	5 (2.8%)	1 (3.6%)	
Ampullary cancer	2 (1.1%)	0 (0%)	
Bile duct invasion from other malignancies	0 (0%)	2 (7.1%)	
Biliary drainage method			<0.001
ERCP	162 (91%)	17 (60.7%)	
PTBD	16 (9%)	11 (39.3%)	
Timing of biliary drainage			0.806
Within 24 h	119 (66.9%)	21 (75%)	
From 24 to 48 h	42 (23.6%)	5 (17.9%)	
After 48 h	17 (9.6%)	2 (7.1%)	
Positive blood culture	31 (17.4%)	19 (67.9%)	<0.001
Laboratory data			
WBC count (/µL)	9655 (7040–11,657)	15,600 (9700–21,440)	<0.001
NLR	8.07 (4.84–14.35)	22.27 (16.25–34.7)	<0.001
CRP (mg/dL)	4.68 (2.28–8.53)	10.15 (5.2–19.27)	0.001
Platelet (×10^3^/µL)	221.33 ± 92.25	158.36 ± 114.12	0.009
PT-INR	0.99 (0.93–1.06)	1.26 (1.08–1.4)	<0.001
Creatinine (mg/dL)	0.79 (0.65–0.98)	1.28 (0.8–1.89)	<0.001
Albumin (g/dL)	3.8 (3.5–4)	3.3 (3–3.63)	0.002
Total bilirubin (mg/dL)	4.73 ± 5.47	5.7 ± 7.71	0.414
Hospital stay (d)	6.5 (4–9)	12 (8–18.25)	0.001
SOFA score	2 (2–3)	4 (4–6)	<0.001

Values shown are means ± SD or median (25th–75th percentiles). SD, standard deviation; ERCP, endoscopic retrograde cholangiopancreatography; PTBD, percutaneous transhepatic biliary drainage; WBC, white blood cell; NLR, neutrophil–lymphocyte ratio; CRP, C-reactive protein; PT-INR, prothrombin time international normalized ratio; SOFA, Sequential Organ Failure Assessment.

**Table 2 medicina-58-00255-t002:** Fifty-four isolated pathogens from blood culture in 50 patients with acute cholangitis.

Pathogen	Blood Culture (*n* = 54)
Gram-negative	
*Escherichia coli*	33 (61.1%)
*Klebsiella* spp.	11 (20.3%)
*Enterobacter* spp.	2 (3.7%)
*Pseudomonas* spp.	1 (1.9%)
*Serratia* spp.	1 (1.9%)
Gram-positive	
*Enterococcus* spp.	5 (9.2%)
*Streptococcus* spp.	1 (1.9%)

Data are described as *n* (%).

**Table 3 medicina-58-00255-t003:** Changes over time in the neutrophil–lymphocyte ratio in the mild/moderate and severe acute cholangitis groups.

Neutrophil–Lymphocyte Ratio	Mild/Moderate Acute Cholangitis (*n* = 178)	Severe Acute Cholangitis (*n* = 28)	*p-*Value
NLR on admission	8.07 (4.84–14.35)	22.27 (16.25–34.7)	<0.001
NLR on day 1	5.38 (3.25–8.27)	12.11 (9.94–18.66)	<0.001
NLR on day 2	3.48 (2.25–5.55)	9.63 (5.89–12.81)	<0.001

NLR, neutrophil–lymphocyte ratio. Values are median NLR (25th–75th percentiles).

**Table 4 medicina-58-00255-t004:** Sequential changes in the neutrophil–lymphocyte ratio of acute cholangitis with and without shock groups.

Neutrophil–Lymphocyte Ratio	Shock (−) (*n* = 196)	Shock (+) (*n* = 10)	*p*-Value
NLR on admission	8.66 (5.37–16.5)	19.27 (15.55–32.9)	0.02
NLR on day 1	5.69 (3.37–9.83)	13.75 (10.97–21.65)	<0.001
NLR on day 2	3.76 (2.38–6.13)	12.57 (6.87–19.56)	0.018

NLR, neutrophil–lymphocyte ratio. Values are median NLR (25th–75th percentiles).

## Data Availability

The datasets generated for this study are available on request to the corresponding author.

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
