# Peer review of "Clinical Significance of the Neutrophil–Lymphocyte Ratio as an Early Predictive Marker for Adverse Outcomes in Patients with Acute Cholangitis"

_medicina, 2022, doi:10.3390/medicina58020255_

Round 1

Reviewer 1 Report

Thank you for this retrospective study on the value of the neutrophil-lymphocyte ratio (NLR) as a predictor of adverse outcomes in patients with acute cholangitis.

This is a well-described retrospective analysis  well written. I would like to raise some minor points.

General:

#1 The statistics have been appropriately analyzed, though the calculation of the power of study is missing. The discussion could mention the appropriate sample size for adequate powering of the study that the authors mention was not met.

Minor:

#1 A previous Japanese study (J Hepatobiliary Pancreat Sci. 2017;24:81) reported that low IL-7 level was also associated with severe cholangitis and blood culture positivity. I suggest to cite this article in the introduction and discussion section.

#2 The methods section of the manuscript needs further details regarding the definition of early complication associated with ERCP or PTBD.

#3 Please add the name of the company manufacturing the CBC analyzer that can measure the delta neutrophil index for the reader’s information in the discussion section.

Author Response

Thank you for this retrospective study on the value of the neutrophil-lymphocyte ratio (NLR) as a predictor of adverse outcomes in patients with acute cholangitis.

This is a well-described retrospective analysis well written. I would like to raise some minor points.

General:

#1 The statistics have been appropriately analyzed, though the calculation of the power of study is missing. The discussion could mention the appropriate sample size for adequate powering of the study that the authors mention was not met.

A) Thank you very much for your comments. A sample size of 186 patients in each group will be required to achieve a power of 80% with a confidence level of 5% to confirm our results when we assumed a 7.5% incidence rate of mild/moderate acute cholangitis and a 1.5% incidence of severe acute cholangitis among patients with gallstone disease. We have mentioned the calculation of sample size in the discussion section as follows:

“Furthermore, a sample size of 186 patients in each group will be required to achieve a power of 80% with a confidence level of 5% to confirm these results when we assumed a 7.5% incidence rate of mild/moderate acute cholangitis and a 1.5% incidence of severe acute cholangitis among patients with gallstone disease. So, the study is underpowered in terms of drawing definitive conclusions on the association between the NLR and severe acute cholangitis.”

Minor:

#1 A previous Japanese study (J Hepatobiliary Pancreat Sci. 2017;24:81) reported that low IL-7 level was also associated with severe cholangitis and blood culture positivity. I suggest to cite this article in the introduction and discussion section.

A) Thank you for the comment. We have cited the study by Suwa et al. that has investigated the association between IL-7 and severe acute cholangitis in the introduction and discussion section.

#2 The methods section of the manuscript needs further details regarding the definition of early complication associated with ERCP or PTBD.

A) Early complication was defined as a complication that occurred within 30 days of the ERCP or PTBD. We added the definition of early complication in the method section.

#3 Please add the name of the company manufacturing the CBC analyzer that can measure the delta neutrophil index for the reader’s information in the discussion section.

A) We have added the company name (Siemens) in the discussion section.

Reviewer 2 Report

In this very interesting manuscript, Lee et al. describe the role of neutrophil to lymphocyte ratio as an early and simple predictive marker for adverse outcomes in patients with acute cholangitis admitted to a single South Korean Hospital. The manuscript is well written and, in my opinion, despite of its limitations. Please see bellow some suggestions to improve the manuscript: 1) In this study, the overall mortality rate was lower than the observed in previous studies of acute cholangitis. Authors should discuss this finding as severity is linked to NLR. 2) It would be interesting to include SOFA score performance in the comparisons. 3) Did the etiology (Choledocholithiasis x non-Choledocholithiasis) impact the performance of NLR? 4) Did the authors exclude liver transplant recipients at enrollment? According to Beliaev et al. Diagnostic inflammatory markers of acute cholangitis in liver transplant recipients. ANZ J Surg. 2021, the performance of NLR in this population is not so good. This information should be included in methods/discussion (last paragraph, last sentence). 5) Please, include which pathogens were isolated in blood cultures. Did the type of pathogen (gram positive or negative) interfere with NLR? 6) What was the distribution of patients with missing data? Between severe and non-severe cholangitis? Missing data should be included as a limitation.

Author Response

In this very interesting manuscript, Lee et al. describe the role of neutrophil to lymphocyte ratio as an early and simple predictive marker for adverse outcomes in patients with acute cholangitis admitted to a single South Korean Hospital. The manuscript is well written and, in my opinion, despite of its limitations. Please see below some suggestions to improve the manuscript.

A) Thank you very much for reviewing this manuscript. We have read your comments carefully and tried our best to address them one by one. We hope that the manuscript has been improved after this revision.

1) In this study, the overall mortality rate was lower than the observed in previous studies of acute cholangitis. Authors should discuss this finding as severity is linked to NLR.

A) Thank you very much for your comments. This study showed that mortality was 1.5%, which is very low compared to the 5%-10% in other studies but is similar to those of a Korean study (Hepatobiliary Pancreat Dis Int 2016;15:619-625) performing early ERCP. The causes of the favorable outcomes may be due to performing early biliary drainage within 48 h. We treated 187 out of 206 patients (90.7%) with early ERCP or PTBD. We have discussed this finding in the Discussion section as follows:

“This study showed that mortality was 1.5%, which is low compared to the 5-10% in other studies but is similar to those of a Korean study by Park et al performing early ERCP. The causes of the good outcomes may be due to conducting early biliary drainage within 48 h in most patients with acute cholangitis. We treated 187 of 206 patients (90.7%) with early ERCP or PTBD.”

2) It would be interesting to include SOFA score performance in the comparisons.

A) Thanks for the comment. We have included the performance of SOFA score with ROC curve (Fig. 5) in the Results section 3.1 and 3.4. SOFA score showed good performance in discriminating severe acute cholangitis, shock and positive blood culture. The AUCs for predicting severe cholangitis, shock, and positive blood culture were 0.87 (95% CI, 0.82-0.91, p < 0.001), 0.93 (95% CI, 0.88-0.96, p < 0.001) and 0.72 (95% CI, 0.65-0.78, p < 0.001), respectively.

3) Did the etiology (Choledocholithiasis x non-Choledocholithiasis) impact the performance of NLR?

A) Thanks for your comment. The choledocholithiasis group (n =168) had similar NLR (14.73 vs. 11.82, p = 0.285) compared to the non-choledocholithiasis group (n = 38).

4) Did the authors exclude liver transplant recipients at enrollment? According to Beliaev et al. Diagnostic inflammatory markers of acute cholangitis in liver transplant recipients. ANZ J Surg. 2021, the performance of NLR in this population is not so good. This information should be included in methods/discussion (last paragraph, last sentence).

A) Liver transplant recipients were not included because liver transplantation is not performed in our hospital. We included this information in the Methods section (lines 92-94).

5) Please, include which pathogens were isolated in blood cultures. Did the type of pathogen (gram positive or negative) interfere with NLR?

A) Thank you for your nice recommendation. Isolated pathogens from blood culture are shown in Table 2. The baseline NLR between Gram-positive and Gram-negative bacterial groups was not different (median, 18.32 vs. 23.06, p = 0.718).

6) What was the distribution of patients with missing data? Between severe and non-severe cholangitis? Missing data should be included as a limitation.

A) There were 17 patients in non-severe cholangitis and 3 patients in severe cholangitis among 20 patients with missing data. The distribution of patients with missing data was similar compared to enrolled patients. We described missing data as a limitation in the last paragraph of the Discussion section.

Reviewer 3 Report

Thank you for the opportunity to review the manuscript "clinical Significance of the Neutrophil-Lymphocyte Ratio as an Early Predictive Marker for Adverse Outcomes in Patients with Acute Cholangitis" by Lee et al. It is a well-written, understandable and interesting study on the  The authors were able to demonstrate a benefit in the detection of patients with severe septic courses in cholangitis in this purely retrospective study. It would be interesting to see a subsequent prospective study that can show the significance of the NLR in this setting as well. 

Author Response

Thank you for the opportunity to review the manuscript "clinical Significance of the Neutrophil-Lymphocyte Ratio as an Early Predictive Marker for Adverse Outcomes in Patients with Acute Cholangitis" by Lee et al. It is a well-written, understandable and interesting study.  The authors were able to demonstrate a benefit in the detection of patients with severe septic courses in cholangitis in this purely retrospective study. It would be interesting to see a subsequent prospective study that can show the significance of the NLR in this setting as well.

A) We fully agree with your comment that this is a purely retrospective study. We will perform future studies for the validation in a larger sample group.

Reviewer 4 Report

Dear Editor

Thanks for inviting me to review the manuscript medicina-1584325 entitled "Clinical Significance of the Neutrophil-Lymphocyte Ratio as an Early Predictive Marker for Adverse Outcomes in Patients with Acute Cholangitis".

This is a well-designed study with a good methodology and well-written manuscript. I have some minor comments to be addressed by the authors.

  • The novelty/importance of the subject could be further discussed in the introduction section of the manuscript.
  • Please specify the 20 missing data in figure 1. Were they because of death or other complications?
  • In section 3.2., "all three markers were predictive for severe acute cholangitis (P<0.001)" please re-check the P-value and AUC for WBC count. It seems that the WBC curve has crossed the reference line in figure 2.
  • Please also re-check the P-value and AUC of CRP in figure 4. It crossed the reference line too.
  • Please mention the statistical test to check the normality of the quantitative variables.
  • Please revise the conclusion to be based on your results. "In conclusion, NLR on admission has high discriminatory ability for severe disease..." It could be better to say that the NLR had an ability to predict ...... in our study population.

Regards

Reviewer

Author Response

Dear Editor

Thanks for inviting me to review the manuscript medicina-1584325 entitled "Clinical Significance of the Neutrophil-Lymphocyte Ratio as an Early Predictive Marker for Adverse Outcomes in Patients with Acute Cholangitis". This is a well-designed study with a good methodology and well-written manuscript. I have some minor comments to be addressed by the authors.

The novelty/importance of the subject could be further discussed in the introduction section of the manuscript.

A) Thank you very much for your comments. We added the novelty/importance of the subject in the Introduction section (lines 46-47 and lines 52-54).

Please specify the 20 missing data in figure 1. Were they because of death or other complications?

A) Thank you for pointing this out. Missing data were not related to death or other complications. Twenty patients had missing data on baseline NLR (n = 2) and NLR after admission (n = 18). We have specified the 20 missing data in figure 1.

In section 3.2., "all three markers were predictive for severe acute cholangitis (P<0.001)" please re-check the P-value and AUC for WBC count. It seems that the WBC curve has crossed the reference line in figure 2.

A) Thank you for the comment. The P-value and AUC for WBC count in section 3.2. were rechecked by the statistician. The results were not changed.

Please also re-check the P-value and AUC of CRP in figure 4. It crossed the reference line too.

A) Thanks for the comment. The P-value and AUC of CRP in figure 4 were rechecked by the statistician. The results were not changed.

Please mention the statistical test to check the normality of the quantitative variables.

A) Thank you very much for pointing this out. We checked the normality in the distribution of the quantitative data with Kolmogorov-Smirnov test. We have mentioned the method for checking the normality in the Method section (lines 126-127).

Please revise the conclusion to be based on your results. "In conclusion, NLR on admission has high discriminatory ability for severe disease..." It could be better to say that the NLR had an ability to predict ...... in our study population.

A) Thank you for your recommendation. We modified the conclusion as you recommended.